# Synergy between Auranofin and Celecoxib against Colon Cancer In Vitro and In Vivo through a Novel Redox-Mediated Mechanism

**DOI:** 10.3390/cancers11070931

**Published:** 2019-07-03

**Authors:** Yi Han, Ping Chen, Yanyu Zhang, Wenhua Lu, Wenwen Ding, Yao Luo, Shijun Wen, Ruihua Xu, Panpan Liu, Peng Huang

**Affiliations:** 1Department of Experimental Therapeutics, Sun Yat-sen University Cancer Center, Guangzhou 510060, China; 2State Key Laboratory of Oncology in South China, Guangzhou 510060, China; 3Collaborative Innovation Center for Cancer Medicine, Guangzhou 510060, China; 4School of Pharmaceutical Sciences, Sun Yat-sen University, 132 Wai huan East Road, Guangzhou 510006, China; 5Department of Medical Oncology, Sun Yat-sen University Cancer Center, Guangzhou 510060, China; 6Metabolic Innovation Cancer, Sun Yat-sen University, Guangzhou 510060, China

**Keywords:** colorectal cancer, drug repurposing, drug combination, auranofin, celecoxib

## Abstract

Recent study suggests that auranofin (AF), a US Food and Drug Administration (FDA)-approved drug for treatment of rheumatoid arthritis, has selective anticancer activity in various experimental models. Its clinical applications in cancer treatment, however, have been hampered due in part to its relatively moderate activity as a single agent. In this study, we performed a high-throughput screening of the FDA-approved drug library for clinical compounds that potentiate the anticancer activity auranofin, and unexpectedly identified an anti-inflammatory drug celecoxib (CE) that potently enhanced the therapeutic activity of AF in vitro and in vivo. Mechanistically, AF/CE combination induced severe oxidative stress that caused ROS-mediated inhibition of hexokinase (HK) and a disturbance of mitochondrial redox homeostasis, resulting in a significant decrease of ATP generation. The CE-induced ROS increase together with AF-medicated inhibition of thioredoxin reductase cause a shift of Trx2 to an oxidized state, leading to degradation of MTCO2 and dysfunction of the electron transport chain. Our study has identified a novel drug combination that effectively eliminates cancer cells in vivo. Since AF and CE are FDA-approved drugs that are currently used in the clinic, it is feasible to translate the findings of this study into clinical applications for cancer treatment.

## 1. Introduction

Colorectal cancer (CRC) is the third most common cancer type and the second leading cause of cancer-related mortality worldwide, with about 1 million individuals being diagnosed and more than 100,000 dying from the disease, annually, all over the world [1,2]. Despite significant advances in early detection and treatment, the long-term overall survival rates of CRC patients have not been significantly improved during the past decades [3]. Chemotherapy is an important treatment approach for patients with CRC. However, CRC patients who progress after receiving two lines of chemotherapy have limited treatment options [4,5,6]. Therefore, there is an unmet need for new drugs that can efficiently eliminate CRC cells in vivo. Recently, repositioning of the established non-cancer drugs for cancer patients has become an important strategy to develop effective cancer drugs due to relatively lower costs and faster drug development, since information on the mechanisms of drug action, formulations, pharmacokinetics, and safety profiles are already available to facilitate their clinical applications [7,8].

Auranofin (AF) is an oral gold-containing compound approved by the US Food and Drug Administration (FDA) for the treatment of rheumatoid arthritis. AF is known to inhibit thioredoxin reductase (TrxR) and exhibits preferential anticancer activity [9,10]. Our previous study showed that AF was able to preferentially killed leukemia cells in vitro and reduced leukemia burden in vivo [10]. We also demonstrated that AF could eliminate cancer stem-like side population cells through inducing ROS accumulation and inhibition of glycolysis in lung cancer model [11]. The promising anticancer activity of AF observed in various experimental cancer models has led to clinical evaluation of this drug for cancer treatment. Currently, AF is in clinical trials for several cancer types, including leukemia, lung cancer and ovarian cancer (https://clinicaltrials.gov/ct2/show/NCT01419691, NCT01747798, NCT01737502). However, the clinical outcomes reported from some of the clinical trials are unsatisfactory due to only moderate and transient therapeutic activity [12]. Thus, it is important to develop new strategies that could enhance the therapeutic activity of AF in vivo.

Proper drug combination is an effective strategy to improve anticancer activity, since the use of multiple drugs provides an effective mean to target multiple cellular abnormalities with a potential to achieve synergistic therapeutic effect, to overcome drug resistance, and to possibly reduce toxic side effects due to dosage reduction for each individual drug in the combination [13,14]. With clinical applications in mind, we used a library of FDA-approved drugs (1280 drugs in total) in a cell-based high-throughput screening (HTS) assay to identify clinical compounds that could potentiate the therapeutic activity of AF in vitro and in vivo, and investigate the underlying mechanisms. 

## 2. Results

### 2.1. High-Throughput Screening for Drugs that Synergize with Auranofin against Cancer Cells In Vitro

Previous studies have shown that the anti-rheumatoid drug auranofin (AF) possesses anticancer activity in various types of cancer cells. However, its clinical application in cancer treatment has been hampered due to unsatisfactory in vivo therapeutic activity. To identify clinical drugs that can potentiate the therapeutic effect of AF, a cell-based high-throughput assay was performed to screen a drug library of 1280 FDA-approved clinical compounds. As shown in Figure 1, we first exposed DLD-1 cells (colon cancer) to various concentration of AF to establish a dose-dependent inhibition curve (Figure 1A) to select a subtoxic concentration (1 μM) of AF, which alone did not cause significant cytotoxicity. We then incubated DLD-1 cells with the FDA drug library in duplicate and with each drug (designated as compound X in Figure 1B) at a concentration of 5 μM). To one set of the duplicate drug panels, AF (1 μM) was added to each well to test the combined effect of AF and compound X. For the second set of the samples, DLD-1 cells were incubated with each compound without AF to evaluate the impact of compound X alone. After 72 h, cell growth inhibition was measured by MTS assay. The combined drug (AF + X) effect on cell growth was subtracted by the respective X effect to calculate the net potentiation (Figure 1B). Any compound that caused a net increase of >60% in cell growth inhibition was considered as a hit, as illustrated in Figure 1C. Using this criterion, 66 compounds were identified as potential hits. These lead compounds were then selected for further test using multiple concentrations (0.03–100 μM) to evaluate their potential synergistic effect with AF. Drug combination indexes (CI values) were calculated using the CalcuSyn software. These processes of selection and validation, led to the identification of celecoxib, an inhibitor of COX-2, that consistently potentiated the anticancer activity of AF.

### 2.2. Auranofin and Celecoxib Synergistically Induce Colorectal Cancer Cell Death In Vitro

Since celecoxib (CE) is known to have cancer prevention activity and the plasma drug concentrations could be achieved at µM range, we decided to further test whether the apparent synergistic effect of CE in combination with AF observed in DLD-1 could be a general phenomenon in colon cancer, we included two additional colorectal cancer cell lines, HCT116 and HT-29, in the quantitative evaluation of multiple concentrations of CE in combination with AF. As shown in Figure 2, combination of CE and AF resulted in a dose-dependent enhancement of inhibition of cell growth in all three colorectal cancer lines (Figure 2A). Notably, the combination index values between CE and AF in these three cell lines are all less than 1 (Figure 2B), indicating a strong synergy between the two drugs.

We then used two additional assays to further confirm the synergistic drug combination effect. Apoptosis assay was performed to test the acute cytotoxic effect of AF and CE. DLD-1, HCT116, and HT-29 cells were treated with AF (1 μM), CE (10 μM), or both for 48 h, followed by double-staining with annexin V/PI and analysis by flow cytometry. As shown in Figure 2C, the cell survival rates in the AF-treated, CE-treated, and (AF+CE)-treated DLD-1 cells were 88.2%, 81.1%, and 30.8%, respectively, indicating a more than additive cytotoxic effect. Similar synergistic cytotoxicity was also observed in HCT116 and HT-29 cells (Figure 2C and Appendix A). To evaluate the long-term impact of CE and AF on cell renewal and proliferation, a colony formation assay was performed. As shown in Figure 2D and Appendix A, there was a synergy between the two drugs in their inhibition of colony formation in all three cell lines.

Interestingly, the combination of CE and AF exerted only a minimal cytotoxic effect on human normal colon epithelial cells (CCD841) and human normal colon fibroblast cells (CCD112), suggesting a preferential killing of cancer cells by the drug combination (Appendix A).

### 2.3. Combination of AF and CE Induced a Severe Energy Crisis in Cancer Cells

To explore the possible mechanisms by which CE potentiated the anticancer activity of AF, we first evaluated the potential role of COX-2 in affecting the cellular sensitivity. Short interfering RNA (siRNA) was used to specifically knockdown the expression of COX-2 in DLD-1 cancer cells, and their sensitivity to AF and CE was then tested. As shown in Appendix A, the knockdown of COX-2 expression by two different siRNAs (si-1 and si-2) did not affect the cellular sensitivity to AF, CE, or their combination, suggesting that inhibition of COX-2 by CE might not play a significant role in enhancing cellular sensitivity to AF. This negative result prompted us to explore other mechanisms. 

Based on our previous finding that AF could inhibit HK activity and reduce cellular ATP at a relatively high concentration (6 μM) [11], we decided to test if AF and CE combination could synergistically disrupt energy metabolism in colon cancer cells. Since mammalian cells produce ATP via oxidative phosphorylation in the mitochondria and glycolysis in the cytosol [15,16], and cancer cells are known to rewire their metabolism to meet energetic and anabolic needs [17,18], we thus used Seahorse XFe24 extracellular flux analyzer to measure oxygen consumption rate (OCR) and extra cellular acidification rate (ECAR) as key indicators of oxidative phosphorylation and glycolysis, respectively. As shown in Figure 3A, treatment of DLD-1 cells with 1 μM AF alone led to a moderate inhibition of ECAR and OCR, whereas 10 μM CE alone did not caused any significant inhibition of ECAR or OCR. Surprisingly, combination of the same concentrations of AF (1 μM) and CE (10 μM) resulted in a severe inhibition of ECAR and OCR. This suppression of energy metabolism led to a severe depletion of cellular ATP (Figure 3B). Similar patterns of metabolic inhibition and ATP depletion were also observed in HCT116 cells (Figure 3C,D) and in HT-29 cells (Figure 3E,F). Importantly, the severe ATP depletion was detected before the occurrence of cell death. As shown in Figure 3G and Appendix A, the vast majority of cells remain intact at 24 h after treatment with AF + CE, while cellular ATP was depleted by more than 90% at this time point in all three cell lines (Figure 3B,D,F). These results together suggest that induction of energy crisis by the drug combination is the approximal event that occurred before cell death.

### 2.4. Combination of CE and AF Caused Inhibition of Hexokinase Enzyme Activity

Based on our previous observation that AF could inhibit hexokinase (HK) enzyme activity [11], we tested if the combination of AF and CE could lead to more inhibition of HK as a possible explanation for the observed potent inhibition of glycolysis by the drug combination. Colon cancer cell lines were incubated with AF (1 μM), CE (10 μM), or their combination for 24 h, and cellular protein extracts were used for analysis of hexokinase activity. As shown in Figure 4, the relatively low concentrations of AF (1 μM) or CE (10 μM) alone caused only slight/modest inhibition of HK enzyme activity, and their combination potently inhibited HK activity in all three cell lines tested (Figure 4A,C,E). The inhibition of HK was associated with a decrease in glucose uptake and lactate production (Figure 4B,D,F). These results suggested that AF/CE combination suppressed glycolysis via inhibition of hexokinase enzyme activity, possibly due to ROS-mediated damage to HK protein (see below).

### 2.5. AF and CE Combination Induced Severe ROS Stress Leading to Protein Oxidation and Dysfunction of Mitochondrial Electron Transport Chain

We next investigated how the combination of AF and CE affected mitochondrial oxidative phosphorylation. It is known that mitochondrial proteins could be affected by redox status [19,20]. It is also known that AF could inhibit thioredoxin reductase (TrxR) and thus affect the conversion of oxidized thioredoxins (Trx) to the reduced form which, in turn, impact the redox states of other proteins [21,22]. Thus, we first examined the redox status of mitochondrial thioredoxin (Trx2) after drug treatment, using “redox Western blotting” to distinguish the oxidized Trx2 from its reduced form as illustrated in Figure 5A. The results showed that under the normal cell culture conditions (without drug treatment), Trx2 was mainly kept in the reduced state Trx2 (Figure 5B, upper band). Addition of exogenous H_2_O_2_ caused an oxidation of Trx2, which appeared as the lower band. Importantly, treatment of cells with a combination of AF and CE led to an almost complete oxidation of Trx2 in all three cell lines (DLD-1, HCT116, HT-29), whereas either drug alone was insufficient to induce a major shift of Trx2 redox status (Figure 5B). Consistently, analysis of cellular ROS showed that AF or CE alone caused relatively moderate increase of cellular ROS, whereas the combination AF and CE induced a marked increase of ROS accumulation (Figure 5C, in log-scale and Appendix A).

The mitochondrial TCA cycle and the electron transport chain are two main components that determine the mitochondrial energy metabolism. Since mitochondrial aconitate hydratase (ACO2) is an TCA enzyme containing iron–sulfur clusters known to be sensitive to ROS stress [23], we first test the impact of AF and CE on this molecule. Unexpectedly, all drug treatment conditions including AF + CE combination did not cause any significant change in ACO2 protein (Figure 5D). We then evaluated the impact of AF and CE on explore the electron transport chain (ETC) components. There are five complexes (I–V) in the ETC system, and each complex contains various protein subunits. We thus assessed the impact of AF and CE treatment on the five commonly used protein components (CI-NDUFB8, CII-SDHB, CIII-UQCRC2, CIV-MTCO2 and CV-ATP5A) of the five ETC complexes. Interestingly, the mitochondrially encoded cytochrome c oxidase II (MTCO2, a protein component of Complex IV) decreased significantly in the AF+CE combination group (Figure 5E and Appendix A), whereas AF or CE alone did not induce such change. The selective decrease in MTCO2 was observed in all three cell lines tested (Figure 5E). There was no significant change in other respiratory complexes (Appendix A). Analysis of mRNA expression by RT-PCR showed that there was no significant changes in the MTCO2 mRNA level (Figure 5F), suggesting that the change in MTCO2 protein was likely due to alteration in the protein stability.

Since it has been reported that Akt is often activated via phosphorylation as a compensatory mechanism in response to mitochondrial dysfunction [24,25], we then examined the Akt phosphorylation status at S473 when cells were treated with AF, CE, or their combination. Surprisingly, Akt phosphorylation at S473 decreased significantly in cells treated with AF/CE combination (Figure 5E), indicating that the Akt compensatory mechanism in response to mitochondrial inhibition in cancer cells was somehow disabled by the drug combination.

### 2.6. Auranofin and Celecoxib Exhibit Synergistic Therapeutic Effect In Vivo

Based on the synergistic effect of AF and CE in vitro, we further evaluated the potential therapeutic activity of AF and CE in vivo, using mice bearing DLD-1 colon cancer xenografts as a tumor model. Six groups of mice were treated as follows: (1) Solvent control (olive oil); (2) AF 10 mg/kg; (3) CE 20 mg/kg; (4) CE 60 mg/kg; (5) AF 10 mg/kg + CE 20 mg/kg; (6) AF 10 mg/kg + CE 60 mg/kg. All treatments were given via oral administration (P.O.). This treatment protocol was well-tolerated, and the mice appeared normal, physically active, and without significant weight loss (Figure 6A). The mice in group 6 exhibited some moderate decrease in body weights during the first 3–4 weeks of treatment, and eventually regained the body weights. As shown in Figure 6B–D, each drug as a single agent exhibited moderate therapeutic activity, whereas the combination of AF and CE produced significantly better therapeutic outcome. At the end of the experiment, the mean tumor weight in group 5 was 0.321 g, significantly less than that of 0.986 g in the vehicle-treated control group (*p* < 0.001); the mean tumor weight in AF-treated and CE (20 mg/kg)-treated groups were 0.739 g and 0.853 g, respectively. The mean tumor weight in group 6 was 0.168 g, significantly less than that of vehicle-treated control group (*p* < 0.001). It is interesting to note that CE at the dosage of 60 mg/kg also showed significant therapeutic activity and reduced the tumor weights to 0.481 g (*p* = 0.0013). When the protein extracts of the tumor tissues were analyzed by Western blotting, we observed a decrease in MTCO2 level in the drug-treated group, with the drug combination groups showing the most significant reduction of MTCO2 protein (Figure 6E). These data were consistent with that observed in vitro, suggesting similar mechanisms of drug action in vitro and in vivo.

## 3. Discussion

Repurposing existing drugs for new therapeutic indications may not only have the advantage of saving time and costs, but also provide opportunities to gain new understanding of the mechanisms of drug action and the biology of diseases. Auranofin is an antirheumatic drug that has been used in the clinic for many years. This compound has attracted increasing attention in recent years, and is considered as a potential anticancer drug due to its broad anticancer activity as observed in multiple cancer models [8,10,11]. However, its applications to clinical treatment of cancer have been hampered due, in part, to only modest therapeutic activity observed in vivo. For instance, in a clinical trial of AF in patients with chronic lymphocytic leukemia (CLL), although drug-induced oxidative stress and apoptosis were observed, the clinical responses were transient, with limited therapeutic activity [12]. Thus, it is important to identify agents that could be used in clinic to potentiate the therapeutic activity of auranofin for cancer treatment. To this end, we conducted a high throughput screening of a drug library containing 1280 FDA-approved clinical drugs and identified celecoxib, that could enhance the anticancer activity of AF in vitro and in vivo.

The discovery that CE potentiated the anticancer activity of AF is somewhat surprising, since CE is a nonsteroidal anti-inflammatory drug (NSAID) that selectively inhibits cyclo-oxygenase-2 (COX-2). Although CE could be considered as potential cancer prevention agent due to its anti-inflammatory property, its ability to enhance the killing of cancer cells was not anticipated nor could be reasonably explained by its key mechanism of action (inhibition of COX-2). Indeed, we observed that a specific knockdown of COX-2 expression did not enhance the sensitivity of colon cancer cells to AF, suggesting that COX-2 unlikely plays a significant role in affecting cellular sensitivity to AF (Appendix A). Our study showed that AF and CE synergistically induced an energy crisis, which seemed to be a key event leading to cancer cell death. Interestingly, there are reports that rheumatoid arthritis patients treated with celecoxib exhibited lower risk for colorectal cancer [26,27], suggesting potential anticancer activity of this drug. However, the clinical effect of celecoxib in combination with auranofin still remains to be evaluated.

Suppression of glycolysis through inhibition of hexokinase (HK) and abrogation of mitochondrial oxidative phosphorylation via oxidative damage to respiratory chain seem to be two key mechanisms by which AF and CE induce energy crisis in cancer cells. This is supported by the data in Figure 3, Figure 4 and Figure 5. However, the key issue is how AF and CE could synergistically inhibit HK and disrupt mitochondrial respiration? Based on the data we obtained in the current study, together with the published data by other groups, we proposed that AF, through its ability to inhibit thioredoxin reductase, could compromise the cellular ability to reduce oxidized thioredoxins. In the absence of additional oxidative stress, cells could still maintain redox homeostasis by utilizing the available cellular antioxidants. However, in the presence of additional oxidative stress, such as ROS production induced by CE, the cancer cells could no longer maintain their redox balance, and oxidative damage to the redox-sensitive proteins would then occur. This might explain why the drug combination could impact both the glycolytic enzyme HK and the mitochondrial respiratory chain component MTCO2. It is known that the reduced form of thioredoxins play a key role in the repair of oxidized proteins, and the fact that the AF + CE combination caused a shift of the mitochondrial Trx2 to oxidized form (Figure 5B) supports the hypothesis that AF + CE combination would lead to oxidative damage to mitochondrial redox-sensitive proteins and thus cause dysfunction of the mitochondrial respiratory chain. Consistently, previous studies also suggest that aberrant ROS levels and a change in redox status could affect protein structure and function. Damdimopoulos et al. suggest that Trx2 could interact with certain components of the mitochondrial respiratory chain and might play an important role in the regulation of the mitochondrial membrane potential [28]. Interestingly, this study also showed that overexpression of Trx2 could confer resistance to etoposide, consistent with our finding that oxidation of Trx2 led to increased sensitivity to auranofin. Likewise, it is possible that HK protein might also be modified by ROS and lost it enzyme activity. Due to the high molecular weight of HK, it is technically difficult to use redox Western blotting to analyze its redox status. Although we have shown that Trx2, HK, and MTCO2 might be involved in the synergistic cytotoxicity of AF and CE combination, it would be important to knockdown Trx2, HK, and MTCO2 to gain further insights into their respective roles and mechanisms in future studies.

Another important question is how CE could induce ROS generation. It has been reported that CE might exert its anticancer activity by inhibiting AKT, reducing Bcl-2, or elevating ROS [29,30,31,32,33]. Although we were able to show that CE could induce ROS accumulation in colon cancer cells (Figure 5C), the underlying mechanisms still remain unclear. A recent study by Pritchard et al. suggests that CE could promote ROS generation by disrupting mitochondrial respiration chain [34], although the exact CE target in the mitochondria remained unclear. Nevertheless, the ability of CE to induce ROS accumulation in cancer cells seems to be a consistent finding. The ROS stress induced by CE together with the intrinsic oxidative stress in cancer cells would put tremendous burden on the cellular antioxidant systems. A simultaneous inhibition of thioredoxin reductase by AF would likely cause a collapse of the cellular redox system, leading to dysfunction of multiple redox-sensitive enzymes, inhibition of cellular metabolism and, eventually, cancer cell death. In contrast, normal cells are able to better maintain redox homeostasis due to their low level of basal ROS generation. Such a balanced redox status enables them to better tolerate exogenous ROS stress owing to their high reserved antioxidant capacity, which can be mobilized to prevent the ROS level from reaching the cytotoxic threshold [22].

Glycolysis and oxidative phosphorylation are the two main metabolic pathways that generate ATP for the cells. Our previous study has confirmed that AF can affect glycolysis by blocking hexokinase activity [11]. In the present study, we showed that AF could decrease ECAR in three colorectal cancer cell lines, while its combination with CE severely inhibited glycolysis activity likely through ROS-mediated damage to HK. A loss of glycolytic activity would usually lead to a compensatory upregulation of oxidative phosphorylation to generate ATP to support various cellular function [11,35,36]. However, the combination of AF and CE also inhibits mitochondrial respiration (Figure 3), likely due to severe ROS stress. Such double hits on the two energy metabolic pathways lead to a severe depletion of cellular ATP, which occurred prior to cell death.

The diversity in inherent difference in ROS levels and redox status between non-cancerous and cancer cells provides a biochemical basis to develop redox-based therapeutic strategies against cancer [8,22,37]. Indeed, we observed that all three colon cancer cell lines were highly sensitive to AF + CE combination treatment, whereas such a drug combination caused minimum cytotoxicity in normal colon epithelial cells (CCD841) and normal colon fibroblast cells (CCD112), confirming that this therapeutic strategy could preferentially kill cancer cells. Selective induction of cancer cell death without harming normal cells is key to therapeutic success. We discovered that a combinations AF and CE could produce a synergistic and selective anticancer effect on colorectal cancer cells. In particular, 1 μM AF in combination with 10 μM CE seemed toxic enough against DLD-1 colon cancer cells but appeared nontoxic to normal cells (Appendix A). These results seem clinically relevant since AF concentrations of 1–3 μM in plasma are achievable without obvious side effects in patients or in volunteer subjects who received the recommended dose of 6 mg/day for rheumatoid arthritis [38,39]. Similarly, plasma CE concentrations greater than 10 μM are also achievable in human receiving 400 mg/day, which seem well-tolerated [40,41]. This is also confirmed in mice bearing DLD-1 xenografts, where AF/CE combinations showed higher therapeutic efficacy than single drugs. Since AF and CE are currently used in clinic for other disease indications, it may be feasible to use these two drugs for potential treatment of colon cancer.

## 4. Materials and Methods

### 4.1. Cells and Reagents

All cell lines were purchased from ATCC. DLD-1 was cultured in RPMI1640 medium (Gibco, Waltham, MA, USA), HCT116 and HT-29 cell lines were cultured in McCoy’s 5A (Modified) medium and LOVO was cultured in Ham’s F-12K (Kaighn’s) medium supplemented with 10% fetal bovine serum in a humidified conditions with 5% CO_2_ at 37 °C.

Auranofin and celecoxib were purchased from MedChemExpress (Monmouth Junction, NJ, USA). 4-Acetamido-4′-maleimidylstilbene-2,2′-disulfonic acid was obtained from Thermo Fisher Scientific Inc. (Rockford, IL, USA). DTT and H_2_O_2_ were purchased from Sigma-Aldrich (St. Louis, MO, USA).

### 4.2. Cell Viability Assay

Cell viability was measured using MTS assay, as we previously reported [42]. In brief, 2000 cells were seeded in a 96-well plate and treated with the indicated doses of celecoxib (1–100 µM), auranofin (1 µM), or both for 72 h. The optical density at 490 nm was determined using a Multiskan plate reader (Thermo Scientific, Waltham, MA, USA). The combination index values were calculated using CalcuSyn software (Biosoft, Cambridge, UK). The CI values indicate a synergistic effect when <1, an antagonistic effect when >1, and an additive effect when equal to 1.

### 4.3. Apoptosis Assay

The number of apoptotic cells was analyzed by an annexin V/FITC (BD, Franklin Lakes, NJ, USA) according to the manufacturer’s protocol. Cells were harvested and analyzed on Beckman flow cytometer (Beckman Coulter, Miami, FL, USA).

### 4.4. Real-Time Cell Metabolism Assay

XFe24 Extracellular Flux Analyzer (Seahorse Bioscience, North Billerica, MA, USA) was used for real-time analysis of extracellular acidification rate (ECAR) and oxygen consumption rate (OCR) according to the manufacturer’s user guide. In brief, cells were seeded overnight in a Seahorse 24-well culture microplate at density 4 × 10^4^ per well. For ECAR detection, the medium was changed to Seahorse base medium supplemented with 2 mM glutamine on the day of the assay and incubated for 1 h in a CO_2_-free incubator at 37 °C prior to the assay. Injections of drugs (DMSO, 1 μM AF, 10 μM CE or both), glucose (10 mM), oligomycin (1 μM), and 2-DG (50 mM) were loaded onto ports A, B, C, and D, respectively. For OCR detection, the medium was changed to Seahorse base medium supplemented with 1 mM pyruvate, 2 mM glutamine, and 10 mM glucose on the day of the assay and incubated for 1 h in a CO_2_-free incubator at 37 °C prior to the assay. Injections of drugs, oligomycin (1 μM), FCCP (1 μM), and rotenone/antimycin A (0.5 μM) were loaded onto ports A, B, C, and D respectively. Results were normalized to cell number.

### 4.5. Measurement of Cellular ATP

Cellular ATP concentration was detected using an ATP-based CellTiter-Glo Luminescent Cell Viability Kit (Promega, Madison, WI, USA) according to the manufacturer’s instructions. In brief, 2 × 10^4^ cells were seeded in 96-well plates and cultured overnight. Cells were then treated with AF (1 μmol/L), CE (10 μmol/L), or both for 24 h. After drug treatment, 100 μL CellTiter-Glo reagents was added to each well and rocked for 2 min to lyse cell. The samples were kept at room temperature for another 10 min. The ATP contents were recorded as luminescent signal, using a luminescent plate reader (Thermo Fisher, Varioskan Flash, Waltham, MA, USA).

### 4.6. Hexokinase Enzymatic Activity Assay

HK activity was measured using a hexokinase assay kit (ScienCell Research Laboratories, Carlsbad, CA, USA) according to the assay procedures recommended by the manufacturer. Briefly, cells were treated with AF (1 μmol/L), CE (10 μmol/L) or both for 24 h, and cell extracts were prepared and performed for assay of HK activity using the reagents provided in the assay kit.

### 4.7. Measurement of Cellular Glucose Uptake and Lactate Production

Cells were seeded in 96-well plates and cultured overnight. The culture medium was replaced with fresh medium containing AF (1 μmol/L), CE (10 μmol/L) or both for 24 h, and the culture medium from each well was collected for analysis of glucose and lactate contents using a Biosensor Analyzer (Biology Institute of Shandong Academy of Sciences, Jinan, Shandong, China).

### 4.8. Thioredoxin-2 Redox Analysis

Redox analysis of Trx2 was carried out as previously described [43]. In brief, cells after treatment were washed with ice-cold PBS and collected by acid precipitation using ice-cold trichloroacetic acid (10%) for 30 min at 4 °C. Samples were centrifuged at 13,000× *g* for 10 min and resuspended in ice-cold acetone (100%) and incubated for 30 min at 4 °C. Remove acetone from tube by centrifugation at 12,000× *g* for 10 min and pellet was resuspended in lysis/derivatization buffer (50 mM Tris-HCl (pH 8), 0.1% SDS, and 15 mM AMS) by sonication and incubated for 3 h at room temperature. Nonreducing SDS polyacrylamide (15%) gel electrophoresis was performed to separate the oxidized and reduced Trx2. Western blotting was performed using a rabbit anti-Trx2 polyclonal antibody (1:1000 dilution; abcam, Cambridge, MA, USA) and goat anti-rabbit secondary antibody (1:10,000 dilution; abcam).

### 4.9. Measurement of Cellular ROS

Cells (2 × 10^5^) were seeded in six-well plates overnight and treated with AF (1 μmol/L), CE (10 μmol/L) or both as indicated. Then the cells were incubated with CMH2DCF-DA (1 µmol/L) at 37 °C for 15 min. Cells were washed twice with PBS, and cellular ROS level were measured using a Cytomix FC500 flow cytometer (Beckman Coulter, Fullerton, CA, USA).

### 4.10. Western Blotting

Antibodies for detecting human Trx2, MTCO2, ATP5A, UQCRC2, SDHB, NDUFB8, AKT, AKT (phospho S473) and beta-actin were obtained from Cell Signaling Technology (Danvers, MA, USA) or Abcam (Cambridge, MA, USA). Cells were washed twice with ice-cold PBS and lysed in lysing buffer. The concentration of proteins was normalized using a BCA protein assay (ThermoFisher, Rockford, IL, USA). Protein samples were run on a standard SDS-PAGE and transferred to PVDF membranes. Subsequently, the membranes were blotted with specific primary antibodies overnight at 4 °C. After that, the membranes were incubated with appropriate horseradish peroxidase conjugated secondary antibodies, and the signals were tested by the ECL detection system (ThermoFisher, Rockford, IL, USA).

### 4.11. High-Throughput Screening Assay

The FDA-approved drug library of 1280 drugs (1 mM stock solutions in DMSO, 80 drugs/96-well plate, 16 plates) was purchased from MicroSource Discovery Systems (Gaylordsville, CT, USA). A high-throughput screening assay was performed as previously reported but with slight modifications [44]. Briefly, 90 μL DMSO was added to each well of drugs using Aurora Versa 1100 workstation to make a 100 μM working solution. Furthermore, 2000 cells were seeded in new 96-well plates using Aurora Versa 1100 reagent dispenser and grown overnight at 37 °C, followed by 10 μL of DMSO, AF, drug X, or their combination as indicated, and incubated for 72 h. Then, 20 μL MTS was added to each well and incubated for 3h and the optical density at 490 nm was determined using a Multiskan plate reader (Thermo Scientific, Waltham, MA, USA).

### 4.12. Mouse Experiments

Mouse experiments were reviewed and approved by the Institutional Animals Care and Use Committee of Sun Yat-sen University Cancer Center (L102012016020G). 1.5 × 10^6^ DLD-1 cells were injected subcutaneously into the right flanks of 6-week-old female athymic nude mice. After 8 days, mice with tumors about 100 mm^3^ in size were randomly assigned into six groups, each containing six mice. AF was freshly prepared at 2 mg/mL in olive oil. CE was freshly prepared at 4 mg/mL and 12 mg/mL respectively. Mice were treated once a day by intragastric administration (except Saturday and Sunday) for 30 days either with olive oil (vehicle), AF 10 mg/kg, CE 20 mg/kg, CE 60 mg/kg, or two combinations: AF 10 mg/kg + CE 20 mg/kg or AF 10 mg/kg + CE 60 mg/kg. Tumor sizes were measured with electronic calipers and volumes calculated using the formula: (Length × Width^2^)/2. Body weights of mice were also recorded. At the end of the experiment, mice were sacrificed and their tumors were collected, photographed, and weighed.

### 4.13. Determination of mRNA Expression

RNA isolation was performed according to the manufacturer’s protocol (QIAGEN, Hilden, Germany). Total RNA was utilized for cDNA synthesis using Reverse Transcription System Kit (Promega, Madison, WI, USA) as described in the manufacturer’s instructions. cDNA was used for quantitative Real-Time PCR (Bio-Rad CFX96 real-time PCR detection system, Bio-Rad Laboratories, Richmond, CA, USA) using specific primers in a SYBR green reaction to determine mRNA levels. The following primers were used to determine MTCO2 mRNA level. F: 5′-CTGAACCTACGAGTACACCG-3′, R: 5′-TTAATTCTAGGACGATGG GC-3′.

### 4.14. Statistical Analysis

Data were presented as mean ± SD. The statistical significance of differences was determined using the Student’s *t*-test. *p* value < 0.05 was regarded as statistically significant. GraphPad Prism 7 software (GraphPad Software, Inc, San Diego, CA, USA) was used for calculating these statistics.

## 5. Conclusions

A combination of auranofin and celecoxib has been identified as an effective regimen that is synergistically toxic to colorectal cancer cells. Mechanistically, auranofin and celecoxib together induce severe oxidative stress and cause redox-mediated inhibition of glycolysis and mitochondrial oxidative phosphorylation, leading to energy crisis in cancer cells.

## Figures and Tables

**Figure 1 cancers-11-00931-f001:**
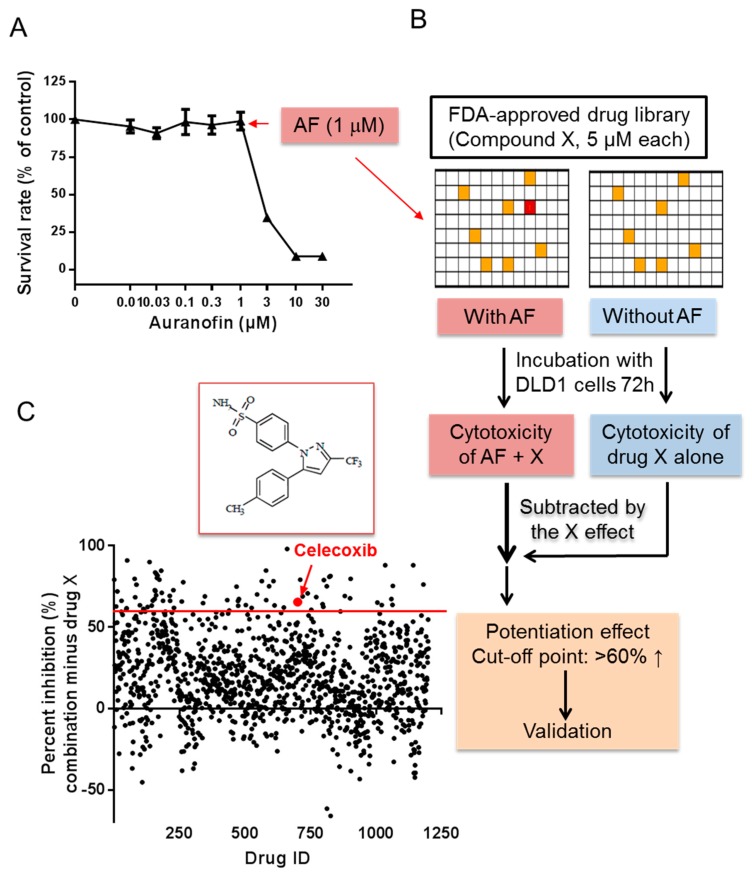
High-throughput screening of drugs that enhance the anticancer efficiency of auranofin (AF). (**A**) Dose–response curves of auranofin treatment for 72 h for DLD-1 cells using MTS assay. (**B**) Scheme showing high-throughput screening for 1280 FDA-approved compounds that enhance the anticancer efficiency of auranofin in DLD-1 cells. (**C**) In this screening, 1 μM AF and 5 μM drug X were used. The inhibition ratio of cell growth increased at least 60% between combination therapy and drug X was considered as a potential target. High-throughput screening (HTS) data of 1280 drugs combined with AF using DLD-1 cells; the red line indicates the threshold (>60 inhibition elevated); the red arrow and dot indicate the final candidate, celecoxib. Data are presented as the mean ± standard deviation (SD) of a triple assay.

**Figure 2 cancers-11-00931-f002:**
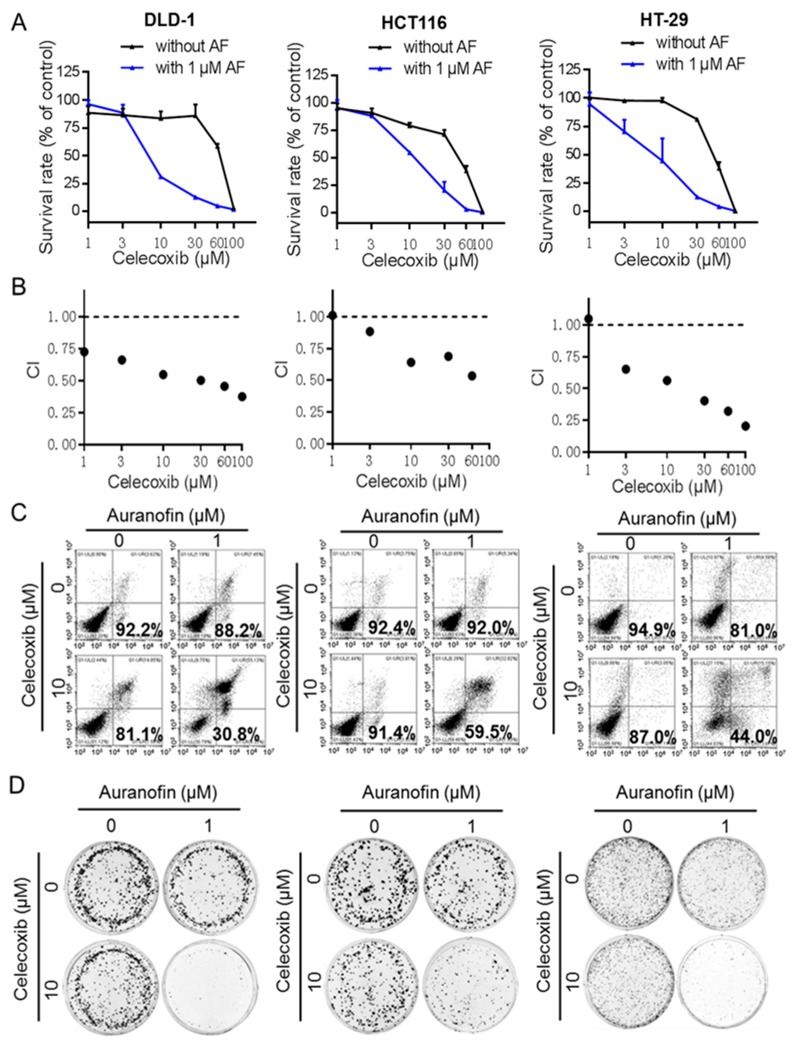
Auranofin and celecoxib synergistically induce colorectal cancer cell death in vitro. (**A**) Dose–response curves of celecoxib with or without 1 μM auranofin for the indicated cells. Data are presented as the mean ± standard deviation (SD) of a triple assay. Cells treated with DMSO alone were used as controls, and their values were set as 1. (**B**) Combination index (CI) of celecoxib and auranofin in the indicated three cell line cells as described in A was analyzed by using CalcuSyn Version 2.0 software (Biosoft). CI > 1 indicates antagonist effect; CI = 1 indicates additive effect; CI < 1 indicates synergistic effect. (**C**) Apoptosis induced by AF (1 μM) and combination with celecoxib (10 μM) for 48 h was detected by annexin V/PI double-staining followed by flow cytometric analysis in the indicated three cell lines. (**D**) The colony formation assay of three cancer cell lines treated with indicated conditions of AF and CE for 2 weeks.

**Figure 3 cancers-11-00931-f003:**
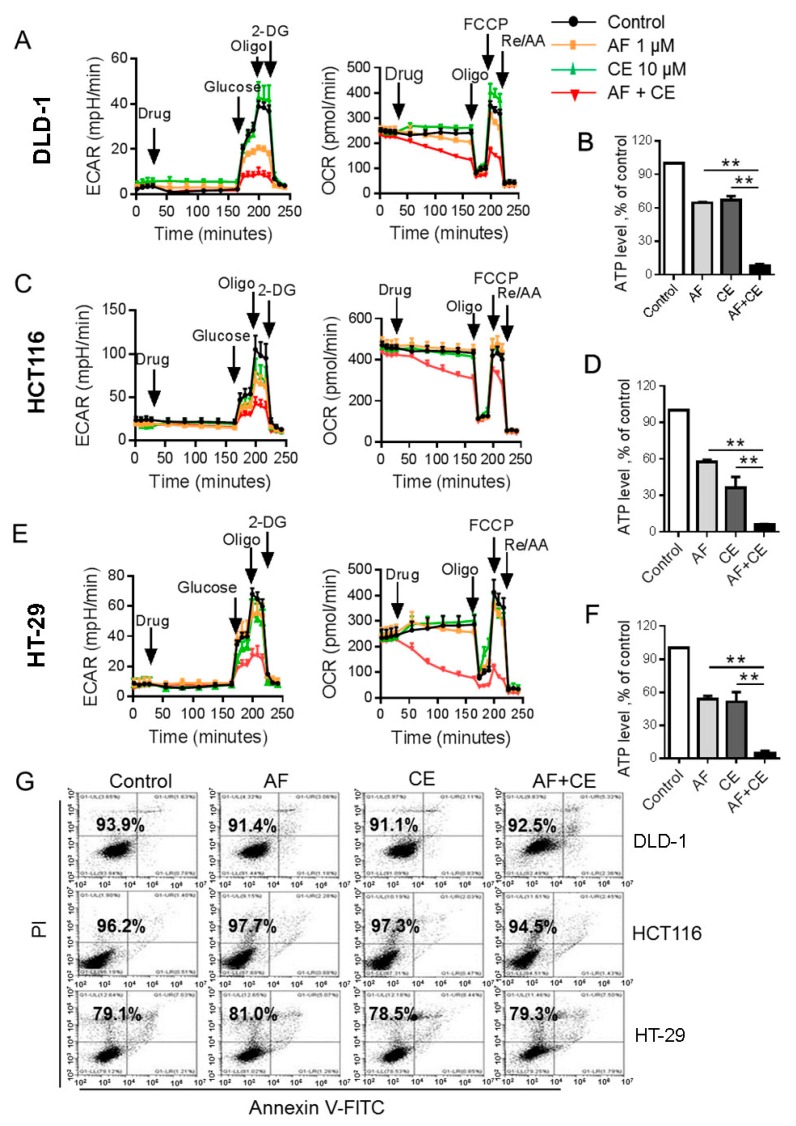
AF/CE combination induced severe energy crisis. (**A**,**C**,**E**) Real-time measurement of ECAR and OCR in indicated conditions using the Seahorse Bioscience Extra Cellular Flux Analyzer. For the glycolysis analysis (left panel), cells were sequentially treated with vehicle or drugs (AF 1 μM, CE 10 μM, and AF 1 μM + CE 10 μM), glucose (10 mM), oligomycin (O, 1 μM), 2-DG (1 μM). For the mitochondrial function analysis (left panel), cells were sequentially treated with vehicle or drugs (AF 1 μM, CE 10 μM, and AF 1 μM + CE 10 μM), oligomycin (O, 1 μM), FCCP (F, 1 μM) and rotenone/antimycin A (R/A, 0.5 μM). (**B**,**D**,**F**) Quantitative bar graph of intracellular ATP from three separate experiments after cells treated with AF (1 μM), CE (10 μM) and their combination (AF + CE) for 24 h. Percent ATP was calculated relative to untreated cells (set to 100%). Each bar represents mean ± SD, *n* = 3 separate experiments; ** *p* < 0.01 (**G**) DLD-1, HCT116 and HT-29 cells were treated with AF (1 μM), CE (10 μM) and their combination for 24 h. Cytotoxicity was determined by annexin V/PI assay.

**Figure 4 cancers-11-00931-f004:**
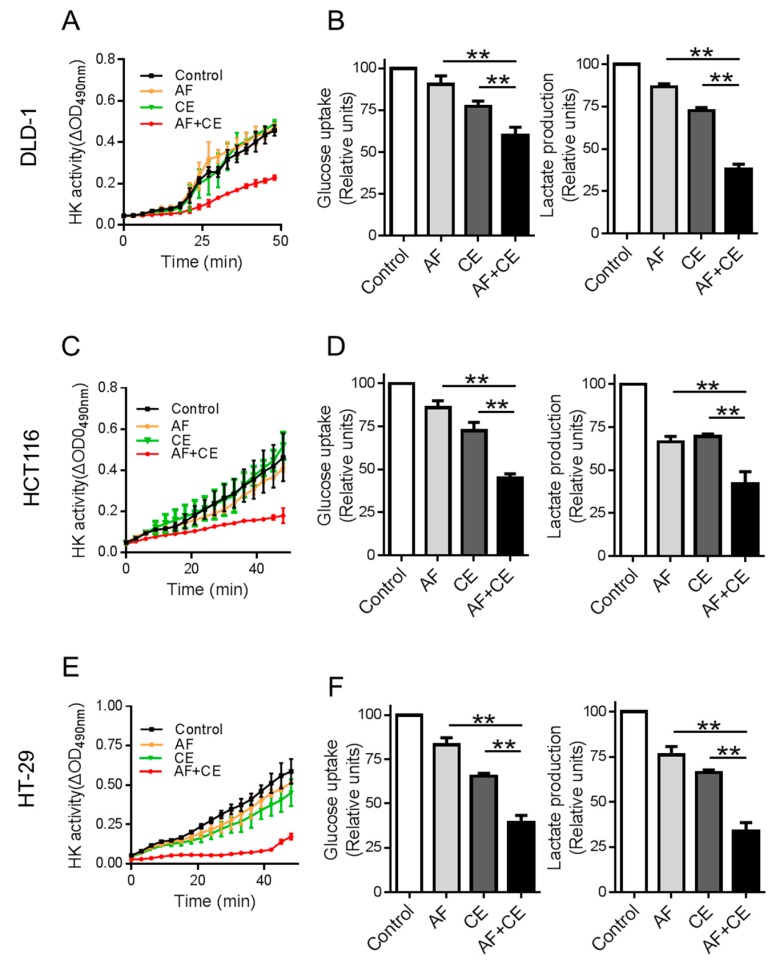
AF/CE combination suppressed glycolysis through hexokinase (HK) inhibition. (**A**,**C**,**E**) Curves of hexokinase activity in cells treated with indicated conditions. Cells were incubated with AF 1 μM, CE 10 μM, and AF 1 μM + CE 10 μM for 24 h, and protein extracts were used to analysis of hexokinase activity. (**B**,**D**,**F**) Inhibition of glucose uptake and lactate production by AF 1 μM, CE 10 μM, and AF 1 μM + CE 10 μM in indicated cells. Three cell lines were treated with the indicated conditions of AF and CE in fresh medium for 24 h. The culture media from each sample was then collected for analysis of glucose and lactate. Each bar represents mean ± SD of three separate measurements, ** *p* < 0.01.

**Figure 5 cancers-11-00931-f005:**
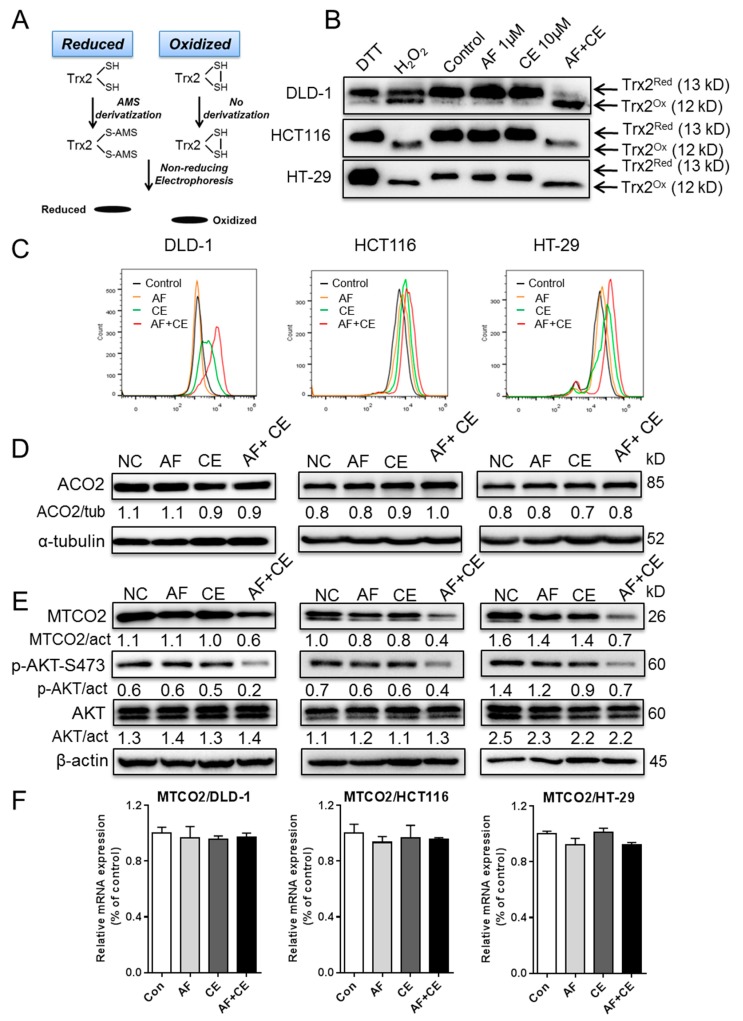
AF/CE combination disturbed redox homeostasis which resulted in disordered electron transport chain. (**A**) Principle of Trx2 redox Western blot assays. DLD-1, HCT116, and HT-29 cells were treated with indicated conditions for 12 h. Treatment with 5 mM H_2_O_2_ for 30 min was used as a positive control (PC). (**B**) Trx2 redox state was evaluated by redox immunoblot. (**C**) Three cell lines were treated with the indicated conditions for 12 h, and cellular ROS levels were measured by flow cytometry analysis after the cells were stained with DCF-DA. (**D**,**E**) DLD-1, HCT116 and HT-29 cells were treated with indicated conditions for 24 h, and cell extracts were analyzed by Western blot for ACO2, MTCO2, p-Akt (ser473) and total Akt protein. (**F**) MTCO2 mRNA level in three cell lines treated with the indicated conditions.

**Figure 6 cancers-11-00931-f006:**
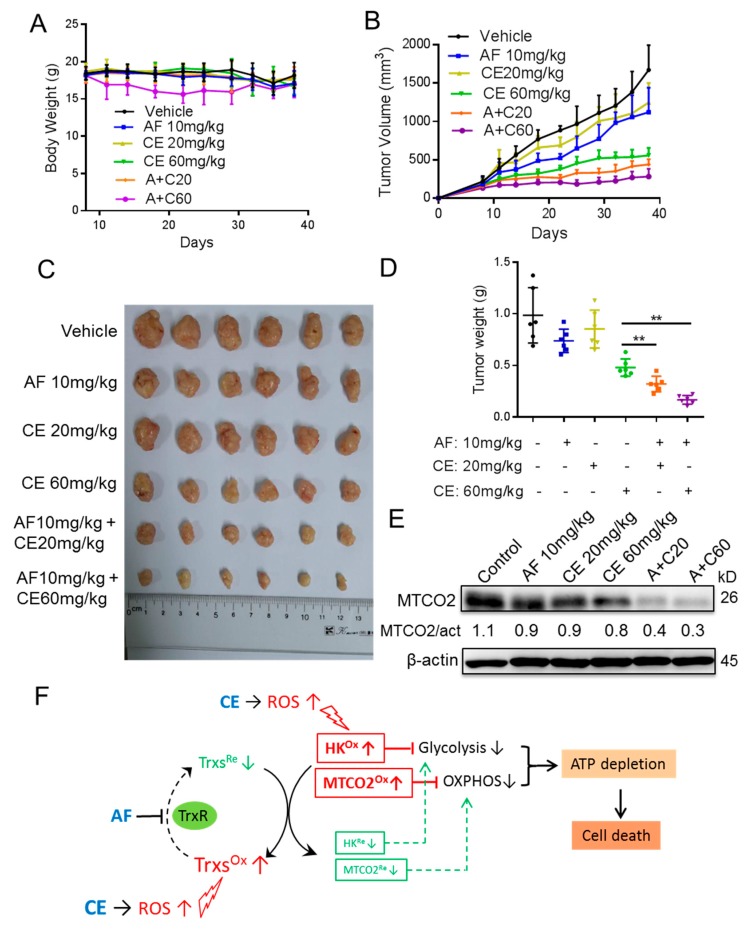
Effect of AF/CE combinational treatment in vivo. Athymic nude mice bearing DLD-1 xenografts were treated with the following drugs via oral injection (P.O.), with olive oil (vehicle), AF 10 mg/kg, CE 20 mg/kg, CE 60 mg/kg, AF 10 mg/kg + CE 20 mg/kg (A + C20), AF 10 mg/kg + CE 60 mg/kg (A + C60). Eight days after inoculation, tumor size and body weight of mice from each group (6 mice per group) were measured twice every week. (**A**) Body weight; (**B**) Tumor growth. Statistical significance of the differences in mean tumor volume between vehicle and indicated groups were analyzed by *t*-test, and all tests were two-sided. (**C**) Gross images and comparison of excised tumor size. (**D**) Tumor weights of different treatment groups at the end of animal experiment (38 days). ** *p* < 0.01 (**E**) MTCO2 protein levels in tumors excised from mice treated with the indicated conditions. (**F**) Schematic model explaining the AF/CE combinational therapy. CE induced ROS increase, which in turn causes oxidation of proteins (Trxs, HK and MTCO2). AF inhibits TrxR, and thus keeps Trxs in oxidized form, which cannot reduce/repair the oxidized proteins (HK, MTCO2) leading to inhibition of both glycolysis and mitorespiration, ATP depletion, and cell death.

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
