# Peer review of "Synergy between Auranofin and Celecoxib against Colon Cancer In Vitro and In Vivo through a Novel Redox-Mediated Mechanism"

_cancers, 2019, doi:10.3390/cancers11070931_

Round 1

Reviewer 1 Report

This paper by Han and coll. entitled “Synergy between auranofin and celecoxib against colon cancer in vitro and in vivo through a novel redox-mediated mechanism” reports a synergistic cytotoxicity of drugs that are already used in clinics (drug repositioning) in colon cancer cells, both in vitro and in vivo, and the underlying mechanism which is energy crisis associated with both glycolysis and mitochondrial respiration inhibition triggered by strong oxidative stress.

The manuscript is well written and the experimental work well performed. However please find below some issues.

Figure 5C: CE is also inducing strong increase in ROS, at least in HT29 cells, thus the sentence “AF and CE alone caused only slight increase of cellular ROS” is not totally correct.

Figure 5E: authors did investigate one protein of each 5 respiratory complexes by Western blotting, but they show only complex IV (Cox2), without mentioning what was observed for the others. They should show them, at least in a supplementary figure.

Figure 6: the in vivo demonstration of synergy is indeed a strong message. One could regret that the mechanism underlying synergy (energy crisis triggered by loss of redox control) was not investigated in vivo.

Page 13 : the last paragraph of the discussion should be deleted.

Discussion: are the RA patients treated with auranofin + anti-inflammatory agent at lower risk of cancer?

Minor issues:

Summary: “Mechanically” should be replaced by “Mechanistically”.

Figure 1A: please add the time point (72h) in the legend.

Discussion: the reference Damdimopoulos #36 is absent.

Author Response

Thank you for the detail review. Please see the attached file.

Reviewer 2 Report

In the manuscript entitled: Synergy between auranofin and celecoxib against colon cancer in vitro and in vivo through a novel redox-mediated mechanism Han and coworkers identified an anti-inflammatory drug celecoxib (CE) that potently enhanced the therapeutic activity of auranofin (AF) in vitro and in vivo. Additionally, they showed that AF/CE combination induced severe oxidative tress and caused ROS-mediated inhibition of hexokinase (HK) and a disturbance of mitochondrial redox homeostasis. Since, AF and CE are FDA-approved drugs currently used in clinic, authors suggest that it is feasible to translate the findings of this study into clinical applications for cancer treatment.

Comments:

1.      Why did authors choose celecoxib? Many other drugs seemed to work better that celecoxib (Fig. 1C).

2.      In case of apoptosis tests, clonogenic tests, flow cytometry studies digital analysis and statistics are missing.

3.      It will be crucial to do knockdown (e.g. siRNA-mediated) experiments to verify involvement of Trx2, HK and MTCO2 in the mechanism proposed by authors. Otherwise, it is just a correlation.

4.      What could be the explanation for lack of Akt compensatory mechanism?

Author Response

(The authors gave the same response as above.)

Reviewer 3 Report

The study “Synergy between auranofin and celecoxib against colon cancer in vitro and in vivo through a novel redox-mediated mechanism” by Han et al. is exciting and a very good piece of work. In this study, the authors have successfully identified the combination of auranofin and celecoxib as an effective regimen that synergistically toxic to colorectal cancer cells. Authors have also shown that auranofin and celecoxib together induce severe oxidative stress and cause redox-mediated inhibition of glycolysis and mitochondrial oxidative phosphorylation, leading to energy crisis in cancer cells. Overall the manuscript is well written, experimental design are appropriate and this study reveals the potential to successful clinical application of these drug combination. Therefore this manuscript could be acceptable with some additional experiments and some minor modifications.

1. Although, authors included the mechanism of action of AF/CE combination in the colon cancer cell, authors need to show the mechanism, why this drug combination is nontoxic to the normal colon epithelial (CCD841) or fibroblast (CCD112) cells.

2. Authors needs to check and clarify the lines 366-368 in the “Discussion” section of the manuscript.

Author Response

(The authors gave the same response as above.)

Round 2

Reviewer 1 Report

The authors addresssed satisfactorily the issues I raised.

Minor remark: error in Fig. S7 for UQCRC2/act.

Reviewer 3 Report

The study “Synergy between auranofin and celecoxib against colon cancer in vitro and in vivo through a novel redox-mediated mechanism” by Han et al. have revised and addressed all the comments. Therefore, this revised manuscript could be accepted by the journal in the present form.